# Metallic Orthodontic Materials Induce Gene Expression and Protein Synthesis of Metallothioneins

**DOI:** 10.3390/ma14081922

**Published:** 2021-04-12

**Authors:** Iwona Ewa Kochanowska, Katarzyna Chojnacka, Edyta Pawlak-Adamska, Marcin Mikulewicz

**Affiliations:** 1Ludwik Hirszfeld Institute of Immunology and Experimental Therapy, Polish Academy of Sciences, R. Weigla 12 Str., 53-114 Wroclaw, Poland; iwona.kochanowska@hirszfeld.pl (I.E.K.); edyta.pawlak@hirszfeld.pl (E.P.-A.); 2Department of Advanced Material Technologies, Faculty of Chemistry, Wroclaw University of Science and Technology, ul. Smoluchowskiego 25, 50-372 Wroclaw, Poland; katarzyna.chojnacka@pwr.edu.pl; 3Department of Dentofacial Orthopaedics and Orthodontics, Division of Facial Abnormalities, Wroclaw Medical University, ul. Krakowska 26, 50-425 Wroclaw, Poland

**Keywords:** orthodontic appliance, nickel exposure, metallothionein expression, reverse transcription

## Abstract

Background: Due to the long-term contact with metallic elements of orthodontic appliances, the potential influence of released metal ions on living organisms and the type of induced changes was investigated. Materials and Methods: Twenty-four young domestic pigs classified in two groups (experimental and control) were chosen as the object of this study. In the experimental group of animals, two metal plates consisting of orthodontic bands representing the mass of orthodontic appliance were mounted on the internal side of the cheek for six months. The liver, lung, and brain samples were taken post mortem from animals of both groups. The gene expression of two isoforms of metallothionein (MT-1 and MT-2) were investigated using the qPCR technique. Protein expression was confirmed by the Western blot and ELISA techniques. Results: The differences in metallothionein concentrations were observed in the lung and brain in the group of experimental animals, but not in the liver. The expression of MT-1 and MT-2 genes in the experimental vs. control group (respectively) was as follows: lung MT-1 1.04 vs. 1.11, MT-2 0.96 vs. 1.05, liver MT-1 0.89 vs. 0.91 vs. 1.12, MT-2 0.91 vs. 1.05, brain MT-1 1.24 vs. 1.20, and MT-2 0.955 vs. 0.945. These results were confirmed by gene activity, which was tested by qPCR. This increased the activity of metallothionein genes in the lungs and brain as a consequence of the release of metal ions into these tissues. The possible effects of detected change in metallothionein-2 gene expression could be the alteration of physiological functions of lung tissue. Conclusions: The effect of long-term exposure to metal orthodontic appliances on metallothioneins gene expression, as well as the induction of protein synthesis was proved.

## 1. Introduction

Nickel is known as an allergenic metal. The frequency of allergic sensitivity of this metal permanently increases. Currently, the percentage of people allergic to nickel is ca. 18% [1,2]. Sensitivity to nickel could be caused by the presence of nickel in clothes, jewellery, dishes, spectacle frames, orthodontic materials or parts of implants [3,4]. Nickel ions absorbed by organism can be widely distributed. Their presence was confirmed in such peripheral tissues, such as hair and nails [5,6]. It means that nickel can penetrate throughout the organism and can pose toxic effects in all tissues. The mechanism of nickel influence on metallothionein synthesis seems to be dependent on transcription factor MTF-1 (Metalothionein Transcription Factor), which is essential for the activation of the metallothionein promoter [7]. Metallothionein plas the major role in detoxification from metals. Metallothionein (MT) is present in four major forms and is synthesized in response to the exposure to toxic metal ions. Thiol groups bind metal ions, thus removing them from the active participation in metabolic reactions. This helps to sustain homeostasis by ion immobilization [8].

While induction of metallothionein expression after exposure to nickel is known and described, studies in the context of metallothionein expression during orthodontic treatment in response to doses of metal ions released from orthodontic appliances is new. It was observed in hepatocytes [9,10,11] and in lungs [12]. The condition and the proper functioning of brain depend on zinc homeostasis in nervous tissue. All four MT major forms are responsible for zinc distribution in the neurons and glial elements and control the redox potential [13]. The proper concentration of zinc is essential for prevention of an extensive number of disorders of the central nervous system. According to the recent findings, oral devices may pose disadvantageous effects, due to the release of metal ions in potentially toxic doses [14].

We tested hypothesis that nickel increases metallothionein synthesis. Also the correlation between synthesis of metallothionein major form 1 and major form 2 was investigated. The potential differences in MT-1 and MT-2 expression, between the control and experimental group of animals—hypothesis 1 (H1), potential differences in MT-1 and MT-2 expression in the same tissue—hypothesis 2 (H2), potential differences in MT-1 and MT-2 expression between tissues—hypothesis 3 (H3)), were tested.

## 2. Materials and Methods

### 2.1. The SS Plates, the Animals and the Tissue Samples

The animal test was performed with the approval (120/2009) of the Ethical Committee of the Wroclaw University of Environmental and Life Sciences. Twenty-four pigs were divided into two groups: the control and the experimental group, each consisting of 12 animals.

Pigs from the experimental group received two plates, implanted on the inner side of both cheeks [15]. The plates were placed on a plastic, polyethylene base, which protected the buccal epithelium from irritation. The plastic elements were placed on the external part of the cheeks. The steel wire was used for clamping plates with plastic bases to the cheeks. The plates were kept on the buccal side of the cheek for 6 months. The pig’s diet, water, and dietary supplements were according to standard conditions. During 6 months of exposure, the weight of pigs increased to 162.3 kg.

The metal elements, the equivalent of orthodontic appliance, were as follows: bands size 37+ (3M Unitek, Monrovia, CA, USA); the steel wire with a diameter of 0.047 inches used for clamping. The composition of the stainless steel alloy (316L) of the experimental plates for pigs was as follows: Fe 65%; Cr 17%; Ni 12%; Mo 2.5%; Mn < 2%; Si < 1%; P < 0.045%; C < 0.03% and S < 0.03% (3M Unitek). The total mass of one plate was 6.2380 g.

The liver, the lung, and the brain tissues were chosen for the examinations. The biopsies of these three tissues were taken *post mortem* from each pig, and frozen at −20 °C. The biopsy was made with ceramic knives to avoid contamination with SS particles/ions. Acid washed vials were used for storage to avoid contamination with trace elements. Each sample mass was about 100 g.

### 2.2. Tissue Homogenates, Total RNA Isolation and Quantification, and Reverse Transcription

Tissue homogenates were obtained from deeply frozen fragments of the liver, lung, and brain tissues, which were next minced to small pieces, and homogenized with a glass homogenizer in PBS (phosphate-buffered saline) on ice, in conditions preventing metallothionein oxidation. Next, the suspensions were subjected to ultrasonication for 30 s and centrifuged for 10 min at 1500× *g* in 4 °C. The supernatants were collected, and the yield of homogenization was estimated in 12% SDS-PAGE, while the protein concentration was determined spectrophotometrically. The supernatants were stored at −20 °C. 

Total RNA was extracted from 200 mg of tissues by using TRIzol Reagent (Darmstadt, Germany) according to the manufacturer’s recommendations. The RNA pellet was dissolved in 20–30 μL of sterile diethylpyrocarbonate-treated Mili-Q water and quantified spectrophotometrically and in agarose gel electrophoresis. RNA samples were stored as an aqueous solution at −70 °C.

Single stranded complementary DNA (cDNA) was synthesized with oligo (dT) primers from 5 μg of total RNA using VerteKit (Novazym, Poznan, Poland) according to the manufacturer’s instruction.

### 2.3. Quantification of Metallothioneins Expression by Real Time PCR

Accumulation of PCR products was measured in Real-Time PCR by using MESA GREEN qPCR MasterMix Plus for SYBR^®^ Assay (Eurogentec, Seraing, Belgium). The sequences of primers for pig metallothionein were chosen in regions of a homological sequence with human metallothionein, as presented in Table 1. The reaction was performed in DNA Engine Opticon^®^ 2 Real-Time Detection System (MJ Research, Bethesda, MD, USA) three times for each probe. Beta-actin was used as a housekeeping gene—β-actin—5′GAGGTAGTCAGTCAGGTCCC3′ and 3′GAAGATCAAGATCATCGC5′. Additionally, product identity was confirmed in agarose (Novazym) gel electrophoresis and visualized by ethidium bromide (Sigma Aldrich, Steinheim, Germany) staining under UV light.

All statistical analyses were performed using the STATISTICA 12 software (StatSoft, Inc., STATISTICA for Windows) using the analysis of variance (ANOVA).

### 2.4. Western Blot

Metallothioneins were detected by Rabbit IgG-anti-porcine MT1/2 (Santa Cruz Biotechnology, Dallas, TX, USA), while the Ag-Ab complexes were detected with mouse monoclonal IgG anty-rabbit-IgG coupled with alkaline phosphatase (Promega, Charbonnières-les-Bains, France).

### 2.5. ELISA

Quantification of MT-2 in liver homogenates was performed using the BlueGene MET2 ELISA Kit (BlueGene Biotech Co., Ltd., Shanghai, China). The determinations were carried out in 3-fold replications of 4-fold diluted samples. The concentration of MT-2 in tested probes was calculated from the standard curve.

### 2.6. Statistical Analysis

The results of the Shapiro-Wilk tests showed the necessity for use of a non-parametric test to analyze statistically significant differences—the Mann-Whitney U test—assuming the significance level *p* < 0.01.

## 3. Results

At the beginning of the experiments, the average weight of pigs was 29.0 ± 1.3 kg, while at the end of the experiment, it was 191.3 ± 10.4 kg. *Post mortem* sampling was performed for the following tissues: liver, lung and brain. The major forms of metallothionein were determined in liver, lung and brain tissues. The homogenates obtained from small pieces of tissues were tested in PAGE-SDS for successful lysis. The protein concentrations in all homogenates were measured and equated before PAGE-SDS and Western blot assays using Rabbit IgG-anti-porcine MT1/2 (Santa Cruz Biotechnology, Dallas, TX, USA). To interpret the significance of the differences between tissues in experimental and control groups of animals for their immunoreactivity, with a clear indication of tissues from tested animals, were discussed. The strongest reaction was observed in all liver samples, from both groups of animals, those from the control and the experimental group. The last two tested tissues (lung and brain) differed from the tissue of liver. The presence of reaction with specific anti-metallothionein—1/2 rabbit antibody appeared only in samples derived from the experimental group of animals. While this reaction was observed in all lung samples, the brain samples reacted to a small extent. The metallothionein was not detected in samples of lung and brain from the control group of animals.

The comparative studies of activity of genes in liver, lung and brain for both types of metallothionein were carried out on the cDNA library synthesized from total RNA isolated from the mentioned tissues. Obtained results were qualified after assessment of the melting curve.

Both forms of metallothionein (MT-1/2) were detected in the livers of all animals from both investigated groups. This does not reflect the possible impact of metal orthodontic appliance on the animal’s organism. Nevertheless, the responses obtained for samples of the lungs and the brain allow us to conclude on its influence on the synthesis metallothioneins.

The influence of metal ions released from orthodontic plate were detected, especially in lung. MT-1 and 2 were determined in lung samples from almost all animals of experimental group and were absent in samples from the control group. The samples of brain tissues only showed reactions in three cases from the experimental group and none from the control.

These data showed the influence of the metal plate on the synthesis of methallotionein in tissues that were not in contact and therefore the mechanism of its action should occur through the metal ions released from the metal plate and to the organism of pigs.

### 3.1. ELISA MT-2 in Liver Samples Determination

The determination of porcine MT-2 was performed in liver homogenates to detect the differences or their lack in the synthesis of this protein based on previous results by Western blots. This test showed the presence of metallothionein in the livers of all animals, but this technique did not allow for quantification of the detected protein. Obtained results showed almost equal MT-2 synthesis in both animal groups. The average MT-2 concentration in livers from the control group was 31 ng/mL, while in the experimental group, it was 29 ng/mL.

### 3.2. Real Time PCR

The statistical analysis concerned the differences in the activity of genes for metallothionein-1 and 2 observed: (i) between the control and the experimental group of animals—hypothesis 1 (H1) (Table 2, Figure 1); (ii) in the same tissue—hypothesis 2 (H2) (Figure 2, Table 3); and between organs i.e., liver, lung, and brain—hypothesis 3 (H3) (Figure 3, Table 4). The null hypothesis (H0)—when *p* > 0.01—was the lack of significant differences between the analyzed samples. The test results are shown in Table 2, Table 3 and Table 4. A box-whisker plot of the ordered *Q_n_* values relative to the analyzed values is included.

Hypothesis 1, which assumed the differences between the experimental and the control groups of animals in gene expression of both metallothioneins, was positively verified. Such difference was demonstrated in the case of MT-2 gene expression in lung tissue. The expression of this gene in the experimental group was reduced (Table 2 and Figure 1).

Hypothesis 2, which assumed the differences in gene expression of both metallothioneins in the same tissue, was also positively verified (Table 3 and Figure 2).

Generally, large differences in the expression of MT-1 and MT-2 genes were noted in the case of two tissues—pulmonary and nervous. In both mentioned cases, the expression of the MT-1 gene significantly exceeded that of the MT-2 gene.

The activity of both metallothionein genes in the lungs of the experimental animals changed and decreased in comparison to the control group, which correlates with the results of H1. However, the MT-1 and MT-2 genes differ from each other significantly in their expression in the nervous tissue. There was no important difference in this parameter between the experimental and the control groups.

Hypothesis 3 assumed the appearance of differences in the expression of MT-1 and MT-2 genes between tissues in the same group of animals. The comparison of MT-1 and MT-2 gene expressions in lung, liver and brain in each group of animals showed the great differences (Table 4 and Figure 3). 

The Western blot technique for simultaneous detection of two metallothioneins MT-1 and MT-2—showed the specific reaction in all three tested tissues, but the significant differences between them were noted (Figure 4).

The largest differences were noted in the expression of the MT-1 gene in all the tissues, but they were almost identical in both groups of animals, which excluded the influence of implanted plate imitating an orthodontic appliance. 

Indeed, the big difference, confirming the hypothesis 3, was shown by the comparison of the lungs and the brain in terms of MT-2 gene expression. The expression in the experimental group decreased in comparison to the control group.

Summarizing, the most significant change in metallothioneins genes expression occurred only in lung tissue and it mainly concerned MT-2 gene expression.

## 4. Discussion

The effect of metallic plate corresponding to orthodontic alloy on the expression and synthesis of two major forms of metallothionein—MT-1 and MT-2—in some tissues was influenced by the degree of specialization of tissue and its regenerative potential. In addition, the role of the organ was important. The following tissues were selected: (i) the liver, the organ of the body’s first line of defense against toxicants, endowed with a regenerative potential; (ii) the lungs, as the tissue of the body with conditioning experience, highly specialized, with the limited regenerative potential; and (iii) the brain, almost devoid of regenerative potential.

The conducted research showed that the lung tissue turned out to be the most sensitive to the potential toxic influence of metal ions. The studies showed differences in the expression of metallothioneins in individual tissues between the experimental and the control animals. According to the obtained results, the expression of both metallothioneins in the experimental and control groups, respectively, was as follows: lung: MT-1 1.04 vs. 1.11, MT-2 0.96 vs. 1.05, liver MT-1 0.89 vs. 0.91 vs. 1.12, MT-2 0.91 vs. 1.05, brain MT-1 1.24 vs. 1.20, and MT-2 0.955 vs. 0.945.The observed decrease in the expression of the MT-2 gene in the lungs was closely related with the increased concentration of this protein. Thus, the initial stimulatory effect, of the metal ions released from the plate, on the expression of the MT-2 gene in the lungs has already been extinguished, but its impact was visible in the elevated amount of MT-2 protein in this tissue.

Metallothionein-2 regulates calcium ions concentration and AKT phosphorylation in smooth muscle cells and MT-2 significantly decreases in asthmatic lung tissue [16]. Metallothionein-2 is related to tissue and cell injury and plays the role in toxicological and chemotherapeutic activity of metal ions and nonmetal electrophiles and oxidants [17]. MT-2 protein levels have been found to be more than 50% lower in asthmatic lung samples than in the control samples [18].

There are published studies showing the MT expression as the prognostic factor or determining marker of different types of tumors. However, there are no constant and repeatable patterns in MT expression that predict cancer. Metallothioneins are upregulated in breast cancer, nasopharyngeal cancer, ovarian cancer, urinary bladder cancer, and melanoma [19,20,21,22,23], while in other cases of cancer (hepatocellular carcinoma, prostate cancer, and papillary thyroid carcinoma), metallothionein expression is downregulated [24,25,26].

The brain tissue seems to be the most resistant to the influence of metal ions released form the plate, because there were almost no changes in MT genes expression and its synthesis was not detected [27]. However, there are evidences on MT participation in neurodegenerative diseases, as well as in brain disorders [27].

The presence of metallothionein was found in all samples of the liver of both groups of animals and no difference in its concentration was confirmed with the ELISA technique. Metallothionein expression could be a useful marker for liver diseases. In acute liver toxicity caused by cadmium, carbon tetrachloride, or acetaminophen, MT plays a protective role. Analysis of MT gene expression in the liver with chronic hepatitis B patients is useful for the evaluation of disease progress. A downregulation of isoform MT-1 in hepatocellular carcinoma was observed in 63% of tumors [28]. Hepatocytes have a high capacity to regenerate, even within few hours after an injury. The regenerating cells require zinc, which is delivered by metallothionein (MT). There are a lot of evidence that MT indicates an essential role for MT in liver cell regeneration [29].

## 5. Conclusions

The results of our study indicate that prolonged exposure to nickel ions released from orthodontic metallic materials, based on tissue homogenates, total RNA isolation and quantification, and reverse transcription, produced observable effects. As a result of the conducted research, it was shown that chronic exposure to metal ions released from orthodontic appliances has an impact on the expression of metallothionein genes and the induction of protein synthesis. The induction of metallothionein, usually resulted in the presence of metals, could be also caused by many other stimuli, such as hormones, cytokines, inflammatory factors, and oxidative stress. The differences in MT genes activity and their protein synthesis indicated in our studies could be the preliminary data for the next studies classifying the safety of orthodontic metal materials in the future.

## Figures and Tables

**Figure 1 materials-14-01922-f001:**
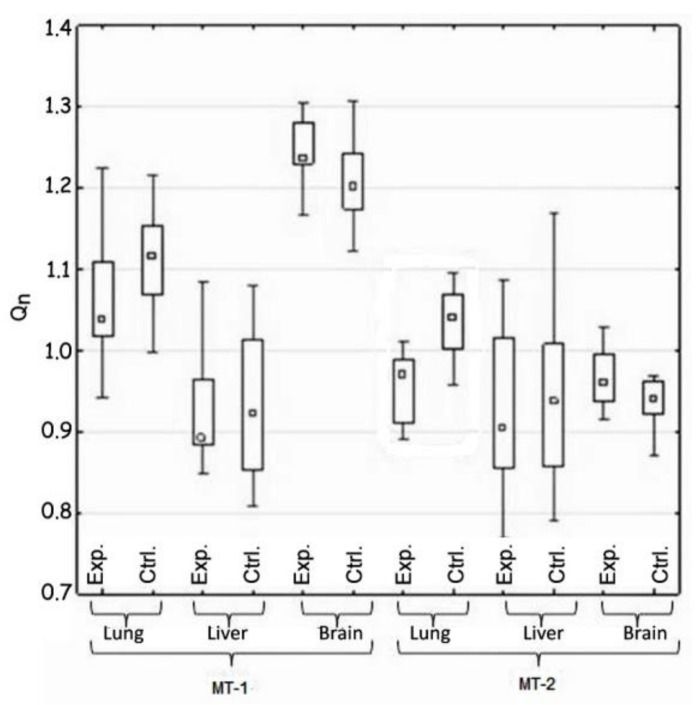
Comparison of the expression of MT-1 and Mt-2 genes in the control and test groups (H1).

**Figure 2 materials-14-01922-f002:**
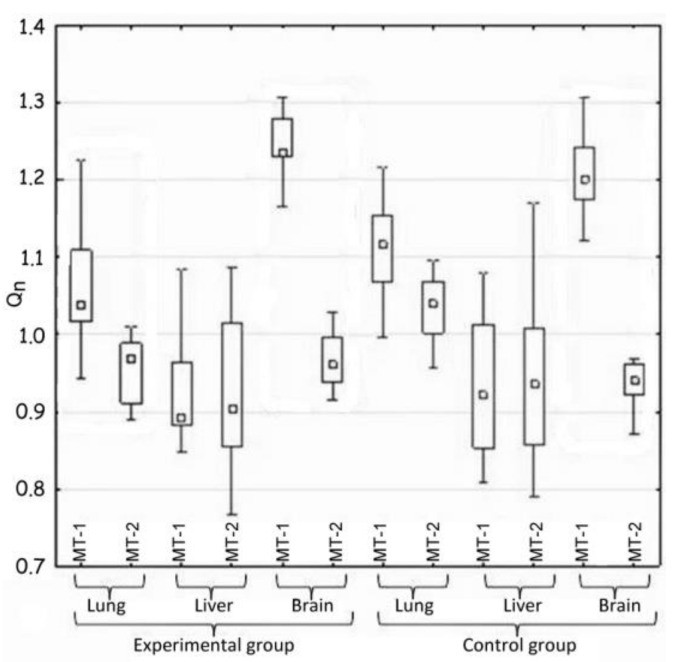
Comparison of the expression of MT-1 and Mt-2 genes in the same tissue (H2).

**Figure 3 materials-14-01922-f003:**
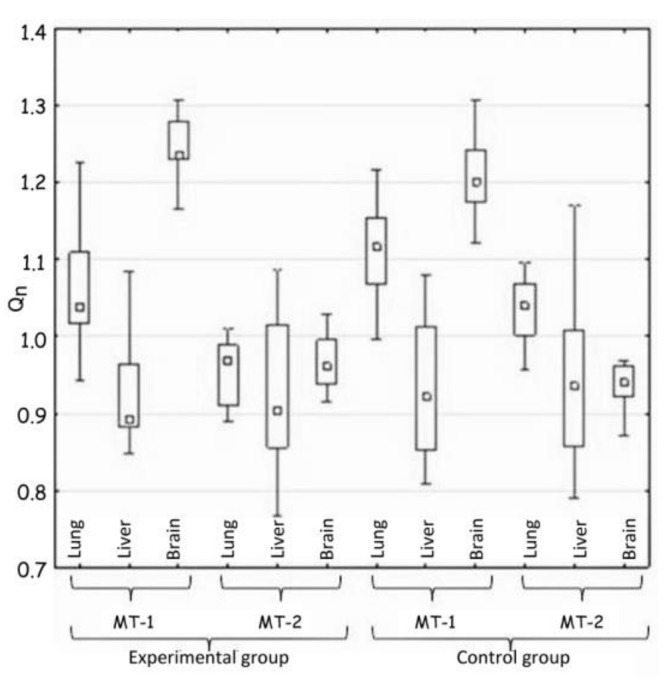
Comparison of tissues in the expression of MT-1 and MT-2 genes (H3).

**Figure 4 materials-14-01922-f004:**
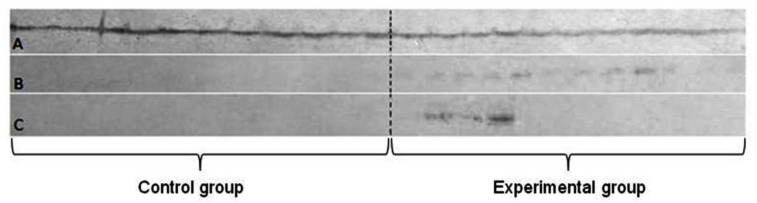
The simultaneous detection of metallothionein 1 and 2 in Western blot technique. Letters stand for: A—liver; B—lung; C—brain.

**Table 1 materials-14-01922-t001:** Primer sequences.

Gene	Sequence	Annealing Temp. [°C]	Loci	Identity [%]	NCBI Gene Accession Number
MT-1	5′ GAGCAGTTGGGGTCCAT 3′	55	67–51	100	NM_001001266.2
	3′ AAGAAGAGCTGCTGCTCC 5′	56	148–155	100	
MT-2	5′ TCTTTGCATTTGCAGGAGCC 3′	58	166–147	100	XM_003355808.2
	3′ ACAAGTGCAGCTGCTGC 5′	55	262–278	100	
β-actin	5′ GAGGTAGTCAGTCAGGTCCC 3′	62	685–666	95	XM_003124280.3
	3′ GAAGATCAAGATCATCGC 5′	52	1093–1110	95	

**Table 2 materials-14-01922-t002:** The Result of the Mann-Whitney U Test for the Significance of Differences between the Control and Experimental Groups in the Expression of MT1 and MT2 Genes (H1) at *p* < 0.01.

			Control Group					
			MT-1			MT-2		
			Lung	Liver	Brain	Lung	Liver	Brain
Experimental group	MT-1	Lung	0.03874					
		Liver		0.62446				
		Brain			0.08176			
	MT-2	Lung				0.0011		
		Liver					0.58649	
		Brain						0.2314

**Table 3 materials-14-01922-t003:** The result of the Mann-Whitney U test for the significance of differences in the expression of MT-1 and MT2 genes in the same tissue (H2) at *p* < 0.01.

			MT-2					
			Experimental Group			Control Group		
			Lung	Liver	Brain	Lung	Liver	Brain
MT-1	Experimental group	Lung	0.00133					
		Liver		0.95662				
		Brain			0.00002			
	Control group	Lung				0.01108		
		Liver					0.68611	
		Brain						0.00003

**Table 4 materials-14-01922-t004:** The result of the Mann-Whitney U test for the significance of differences between tissues in the expression of MT-1and MT2 genes (H3) at *p* < 0.01.

			Experimental Group				Control Group			
			MT-1		MT-2		MT-1		MT-2	
			Liver	Brain	Liver	Brain	Liver	Brain	Liver	Brain
Experimental group	MT-1	Lung	0.00827	0.00004						
		Liver	1	0.00004						
	MT-2	Lung			0.19098	0.66291				
		Liver			1	0.09558				
Control group	MT-1	Lung					0.0071	0.00291		
		Liver					1	0.00015		
	MT-2	Lung							0.0171	0.00291
		Liver							1	0.57674

## Data Availability

The data presented in this study are available upon request from the corresponding author.

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
