# Peer review of "Metallic Orthodontic Materials Induce Gene Expression and Protein Synthesis of Metallothioneins"

_materials, 2021, doi:10.3390/ma14081922_

Round 1

Reviewer 1 Report

This is a very interesting study on the release of metallothioneins from metallic orthodontic devices normally used in the oral cavity.

The topic is certainly of scientific interest, but many criticisms are present in the work.

In particular:

-Abstract section: it is useless to define the pigs as similar to humans but rather to define the two study groups

-In the final part of the abstract section enter the possible clinical effects of the molecular expression from the analyzed devices

- The keywords must be exclusively MeSH terms of the main search engines

-The introduction section presents the biggest problems and needs to be completely rewritten.

First of all, it is necessary to emphasize the possible toxic effects of oral devices in general, aspects that have been highly regarded in the international scientific literature of recent years. In this regard, I recommend that you insert the following scientific work in the reference section, which could be of help to the reader:

Pagano S, Coniglio M, Valenti C, Negri P, Lombardo G, Costanzi E, Cianetti S, Montaseri A, Marinucci L. Biological effects of resin monomers on oral cell populations: descriptive analysis of literature. Eur J Paediatr Dent. 2019 Sep; 20 (3): 224-232. doi: 10.23804 / ejpd.2019.20.03.11. PMID: 31489823.

-The hypotheses of the study should be presented as null hypotheses to be refuted at the end of the study in the light of the results obtained

-What were the selection criteria for the sample size?

- Why was the code of approval of the ethics committee of the reference body not included in the initial part of the study design?

-Inserting the weight before and after the study should not be included in the part of the materials and methods but rather in the results

-The homological sequences must be inserted in a specific table

- When naming companies or products (eg Eurogentec, Sigma Aldrich) also indicate country and product name

- Statistical analysis must be inserted as the final paragraph of the materials and methods and not in the results

-The results section should not be commented on in the light of the hypotheses of the study but simply the results should be listed. This part should be completely rewritten

-The reference section also appears to be revised especially for the number of old works inserted

Reviewer 2 Report

The Authors must see my remarks

It is an interesting article

Round 2

Reviewer 1 Report

I reccomend work acceptation 
